

# Vertical wind profiling from troposphere to the lower mesosphere based on high resolution heterodyne near-infrared spectroradiometry

Alexander V. Rodin[1,2] , Dmitry V. Churbanov[1], Sergei G. Zenevich[1], Artem Yu. Klimchuk[1,3], Vladimir M. Semenov[3], Maxim V. Spiridonov[1,4], Iskander Sh. Gazizov[1].

[1] Moscow Institute of Physics and Technology, Dolgoprudny, Russia
[2] Space Research Institute, Moscow, Russia
[3] Samsung R&D Institute, Moscow, Russia
[4] Prokhorov General Physics Institute, Moscow, Russia

*Correspondence to*: Alexander V. Rodin (alexander.rodin@phystech.edu)

**Abstract.** We propose a new technique of remote wind measurements based on Doppler analysis of a $CO_2$ absorption line in the 1.605 μm overtone band measured in the direct Sun observation geometry. Heterodyne spectroradiometric measurements of the solar radiation passed through the atmosphere provides an unprecedented spectral resolution up to $\lambda/\delta\lambda \sim 10^7\text{-}10^8$ with a signal-to-noise ratio more than 100. The shape of the individual rotational line profile provides unambiguous relationship between offset from the line centre and altitude where a respective part of the line profile is formed. Therefore, an inverse problem may be posed in order to retrieve vertical distribution of wind, with retrievals vertical resolution compromised by a spectral resolution and signal-to-noise ratio of the measurements. A close coincidence between measured and synthetic absorption line is reached, with retrieved wind profile between the surface and 50 km being in a good agreement with reanalysis models. This method may pose an alternative to widely employed lidar and radar techniques.

## 1 Introduction

In spite of the tremendous progress in remote sensing during last decades, atmospheric dynamics still remains hard to assess by direct measurements. To date, the most reliable means of the remote wind field measurement are Doppler radars whose sounding range extends up to the lower thermosphere (Woodman and Guillien, 1974, Hocking, 1997, Hysell et al., 2014). Doppler lidars typically operating in the infrared spectral range are widely used at smaller scales (Eberhard and Schotland, 1980, Bruneau et al., 2015). These methods are based on active sounding of the medium by a powerful source (laser of radar beam), implying significant required resources in mass, size, and power consumption. In contrast with radar and lidar sounding, passive Doppler spectroradiometry has received much less attention in atmospheric studies. Recent developments in the microwave Doppler wind velocimetry have been reported by Newnham et al. (2016). The authors argue that in the stratosphere and lower mesosphere, passive high resolution spectroradiometry in the microwave remain one of the most effective technique of direct wind monitoring. Not only heterodyne spectroscopy provides information about wind field in


the Earth atmosphere, but also in the atmospheres of other planets. Taking advantage of passive observations capabilities of sounding remote objects at arbitrary distances, high resolution heterodyne spectroscopy in the microwave and mid-infrared spectral ranges has provided an opportunity to measure winds in the atmospheres of Venus, Mars, and Titan by means of ground-based telescopes (Kostiuk et al., 1983, Sornig et al., 2012). High resolution spectroscopic studies with traditional

echellé or Fabri-Perot spectrometers in the visible spectral range has been also employed for wind measurements in the atmospheres of Earth and other planets from the ground-based telescope (Widemann et al., 2007) and Earth-orbiting satellite (Killen at al., 2006). However, direct wind measurements beyond the troposphere of the Earth, as well as of other Solar System planets, remain largely peculiar studies rather than well-established technique.

As radiometric Doppler velocimetry requires extraordinary spectral resolution $\lambda/\delta\lambda \sim 10^8$, which could be most effectively

implemented using heterodyne detection technique. However, instruments for atmospheric heterodyne radiometry operating in the microwave and terahertz ranges are typically heavyweight and expensive, and hence their implementations at spaceborne and airborne platforms are limited. Ground-based stations equipped with such instruments are in the limited use because of relatively high cost as well. We propose a technique that allows Doppler measurements in the near-infrared spectral range by means of a compact, lightweight, and simple near-infrared spectroradiometer in the direct Sun observation

mode. Heterodyne method is rarely used for spectroscopic measurements in this range for various reasons. First of all, there is a fundamental limit to the signal level when observing an extended object, as the field of view of any heterodyne radiometer is constrained by a diffraction limit due to antenna theorem (Siegman, 1966). Thus if the Sun is used as a light source for atmospheric sounding, an entrance aperture as small as 0.3 mm in diameter would provide the maximal signal to be put into the heterodyne instrument working at 1.605 μm. On the other hand, the shorter wavelength, the higher photon

energy, and quantum limit of the heterodyne detection sensitivity increases accordingly. For instance, for the wavelength $\lambda$ = 1.605 μm, the quantum limit in terms of noise-equivalent power (NEP) is $\sim 10^{-19}$ W/Hz. With the resolved bandwidth B = 10 MHz and reasonable exposure time $\tau$ up to few minutes, the quantum limit constrains heterodyne detection by a minimal

level of spectral brightness of $p = \dfrac{hc}{\lambda\sqrt{B\tau}} \approx 2\cdot10^{-24} W/\mathrm{Hz}$. Therefore, applications of heterodyne detection in the

shortwave infrared range is limited to the cases where either Sun or artificial non-thermal light sources characterized by high

spectral brightness are used, such as LEDs or lasers.

Several examples of near-infrared heterodyne spectroradiometers utilizing Sun as a radiation source have been recently developed for the purpose of greenhouse gas measurements, including methane and carbon dioxide (Rodin et al., 2014, Wilson et al, 2014, Melroy, 2015, Kurz et al., 2016, Hoffman et al., 2016, Zenevich et al., 2019, Deng et al., 2019). Weidmann et al., (2017) took advantage of the compact design of a fiber-based heterodyne spectroradiometer and proposed a

cubesat-class mission for methane isotopologue measurements. The above mentioned authors focused on the advantage of high resolution spectroscopy in sensitive measurements of absorber amounts and distinguishing isotopes of the same species. In this work we analyze capabilities of this method to fetch spectroscopic information for remote Doppler velocimetry.





## 2 The experiment

### 2.1 Fundamentals of tuneable diode laser heterodyne spectroradiometry

The basic principles of the diode laser heterodyne spectroradiometry (TDLHS) have been presented in detail in previous works, e.g. in Rodin et al. (2014), Kurz et al. (2016), Zenevich et al. (2019). Here we briefly repeat the core information important for Doppler measurements. In contrast with conventional microwave heterodyne spectroradiometry, TDLHS operates with a tunable local oscillator (LO) – a diode laser with distributed feedback (DFB), characterized with a bandwidth of about 2MHz and capable of changing its radiation frequency in a broad spectral range. Typically, two parameters are

employed to control DFB laser radiation frequency – its temperature and injection current. Temperature affects the resonant frequency through the period of an embedded Bragg grating, while pumping current impacts both grating period (through temperature variations according to absorbed power) and refractive index (through charge carrier density). Commercial DFB lasers are supplied with embedded Peltier coolers and thermal sensors, so both parameters could be precisely controlled by external electronics.

We apply the algorithm of LO sweeping with locking to reference absorption line, as described in Rodin et al. (2014). According to this algorithm, temperature controller is set at constant value, while pumping current is modulated by a piecewise linear saw-edged function. Both laser power and wavelength change accordingly. Signal acquired simultaneously in the heterodyne channel is characterized by linear growth of the received power according to changing LO intensity, and by a random heterodyne component:

$$i_{het} = D\left(E_S E_{LO} + E_S^* E_{LO}^*\right)$$

$$= D\mathrm{Re}\int_0^\infty g_{LO}(\omega')d\omega'\int_0^\infty F_S(\omega)\exp\left[-i(\omega-\omega')t + i\varphi_S(t) - i\varphi_{LO}(t)\right]d\omega,$$
(1)

where $D$ is the detector responsivity, $g_{LO}$ and $F_S$ are power spectral densities of the LO and solar light at LO frequency, while $\varphi_{LO}$ and $\varphi_S$ are their respective phases. Power spectrum of the beat signal is proportional to the convolution of power spectra related to the signal $F_S$ and local oscillator $S_{LO}$, respectively:

$$S_{het}(\omega) = 2D\int_0^\infty S_{LO}(v-\omega)d\omega'\int_0^\infty F_S(v)\,dv.$$
(2)

As the analyzed solar radiation is not expected to reveal features narrower than the LO bandwidth, i.e. $F_S$ may be assumed constant within the bandwidth $B$, corresponding to the peak of $S_{LO}$, which in turn may be replaced by delta function, the convolution in the right hand side of Eq. (2) could be reduced to a simple product. Due to the random phase factor, within a narrow frequency range with respect to LO bandpass, $\Delta\omega \le \Delta\omega_{LO}$, heterodyne signal reveals properties of a white noise with the dispersion, proportional to the target spectral density $F_S$:

$$\left\langle i_{het}^2 \right\rangle = 2D\,B\,i_{LO}\cdot F_S(\omega_{LO})$$
(3)



Therefore, to achieve heterodyning of the thermal non-coherent IR radiation, one needs, at first, to subtract a linear component from the signal associated with sweeping the LO power, and then to obtain the dispersion by square detection. Finally, appropriate exposure and data collection provides target spectrum $F_S(\omega)$ with required signal-to-noise ratio.

Taking into account that LO linewidth has an order of 2MHz, the spectral resolution of heterodyne detection is sufficient to

measure Doppler shift of the absorption line in the atmosphere due to air mass motion with velocities greater than 3 m/s. Note that sweeping the LO frequency over spectral range of interest and replacement intermediate frequency (IF) spectral analysis by mere dispersion evaluation implies much simpler instrument design compared to conventional heterodyne spectroradiometers, as complex backend equipment is no longer necessary. Further details of data treatment depend on the optical scheme and should be discussed in the link with description of the experimental setup.

**2.2 Experimental setup and observations**

Detailed description of the experimental setup is presented in Zenevich et al. (2019). Here we shortly repeat the main information important for understanding. A sketch of the experimental laser heterodyne spectroradiometer (LHS) is presented in Figure 1. Solar radiation passed through the atmosphere is focused to the edge of a single mode silica optical fiber with core diameter ~ 10 μm by a lens with focal length $f$ = 50 mm. A chopper installed in front of the fiber allows to

record heterodyne signal with the Sun on and off. Thus the dark signal, dominated with the shot noise, and superposition of the heterodyne and dark signals are measured separately. This technique provides accurate subtraction of the shot noise unaffected by possible system drifts of the instrument. Sunlight entering the fiber has been combined with a LO radiation in a single mode silica coupler with 5:95 coupling ratio. According to antenna theorem (Siegman, 1968), single mode fiber geometric aperture factor ~2.6·10⁻⁸ cm², corresponding to field of view ~ 0.006°, is close to the maximum available for

heterodyne detection expressed by and therefore does not limit the instrument sensitivity (Rodin et al., 2014). An automated SkyWatcher EQ6 PRO SynScan equatorial mount equipped with QHY5-II guide camera was used to track the Sun during observations.

Radiation of the NEL DFB laser serving as a LO is split in 50:50 proportion, with one portion being coupled with solar radiation in the heterodyne channel, and another portion being put into the reference cell. All fiber-optical connections in

both heterodyne and reference channels are made by welding with the exception of two FC/APC angled connectors to LO and to the telescope, to avoid undesired reflection, which may otherwise significantly contaminate the signal with parasite interference pattern. Frequency ramping in the range $6230.2\pm0.35$ cm⁻¹ was provided due to modulation of the pumping current by a chain of pulses, with pulse length of 39.4 ms, dead time of 10.6 ms, and stepwise current growth within a pulse with a step length of 200 μs. The cell filled with $CO_2$ at pressure of 13 Torr is an off-axis resonator with the input and output

mirrors reflectivity $R_{in}$ = 0.9998 and $R_{out}$ = 0.98, respectively, which provides the effective optical path length ~30 m. This path allows for reliable measurement of Doppler-broadened contour of the weak absorption line in the 1.605 μm band by means of the integrated cavity output spectroscopy (ICOS) technique (Moyer et al., 2008). Negative feedback between the



detected line position in the reference cell and diode laser crystal temperature has been organized in order to stabilize LO

frequency ramping and to avoid drifts, with the crystal temperature correction accuracy of $10^{-4}$ K. After coupling solar

radiation with the LO, single mode fiber transmits the signal into Hamamatsu InGaAs p-i-n diode photodetector serving as a

heterodyne mixer. Then the beat signal is analyzed with backend electronics.

After the transimpedance preamplifier, the signal is passed through consecutive low-pass filter, another amplifier and high-

pass filter, that limits its bandwidth to 0.2…10 MHz. Preamplifier circuit is connected to Rohde & Schwarz RTO 1012

digital oscilloscope at two points: after the filter chain and just after transimpedance preamplifier. The latter signal repeats

LO pumping current pulse shape and is utilized for oscilloscope synchronization with the LO ramping cycle. The first is used

for dispersion evaluation and data collection. Oscilloscope translates the signal into digital sequence with the rate of 20 000

points per second, then calculates the dispersion and averages data in a similar way as it has been previously done by Wilson

et al. (2014) and Kurz et al. (2016). The oscilloscope communicates with the PC by Ethernet channel, while the other parts

of the setup, including LO control system and Sun tracker, are connected by USB. Instrument control and data transfer are

implemented based on LabView platform.

As the correct frequency calibration over the whole range of LO ramping is crucial for Doppler analysis of the atmospheric

absorption, we employed another reference channel based on Fabri-Perot etalon. Frequency calibrations were done offline,

as described in Rodin et al. (2014), and utilized for processing and interpretation of atmospheric absorption measurements.

In order to estimate the accuracy of frequency absolute stabilization by the reference cell and relative calibration by Fabri-

Perot etalon, deviations in LO frequency ramping from the prescribed sequence have been tracked along several hours.

These tests have shown RMS deviation of 1.5 MHz in the vicinity of the reference $CO_2$ line, and 6 MHz in the rest of the LO

ramping range.

Several observation sessions have been carried out during the summer 2018 under clear sky conditions. The instrument has

been installed at the top of one of the buildings at MIPT campus at the North of Moscow city, with coordinates 55.929036º

N, 37.521506º E, about 50 m above the sea level. Data were collected for ~10 min, to reach signal-to-noise level of 100.  The

details of observations and some key characteristics of the retrieved spectra are presented in Table 1. Further development of

this technique is possible by implementing of multichannel photodetector configuration. This could significantly decrease

the integration time and increase signal-to-noise ratio.

## 3 Wind profile retrievals

### 3.1 Raw data treatment

Once heterodyne signal is obtained, one needs to subtract corresponding dark signal with the solar channel being closed by a

chopper. The next step is to eliminate baseline slope caused by the modulation of LO power. Since heterodyne signal and

shot noise have the same statistical nature, being proportional to the power absorbed by a photomixer, this slope cannot be

eliminated by dark signal subtraction and requires model approximation of the spectral continuum outside the observed



absorption line. The ratio of heterodyne signal with subtracted dark signal and baseline approximated by square polynomial is the final transmission spectrum of the atmosphere. Consecutive steps of raw data treatment are presented in Figure 2 (a – c). In Figure 2a one could notice baseline distortion in the reference channel caused by the residual interference in the ICOS multipass cell. This distortion may be eliminated by a precise adjustment of the entrance beam. However, this channel is only used for determination of the line peak position, which is not sensitive to the baseline variations.

Further treatment of LHS data implies comparison with a synthetic spectrum of the atmospheric transmission. Since in the narrow field of view of the LHS scattered component is negligible compared to direct solar radiation, atmospheric spectra simulation is straightforward. In addition to the target $CO_2$ line R2 $14^01\leftarrow00^00$ at 6230,22 cm-1, other $CO_2$ lines at 6230,25 cm$^{-1}$, 6230,02 cm$^{-1}$ and 6229,98 cm$^{-1}$ have also been included in calculations. Other atmospheric species, including water vapor, do not affect this spectral range with features that could be distinguished from the spectral continuum. Based on

HITRAN-2016 data (Gordon et al., 2017), the model of the atmospheric transmission spectrum has been constructed. In calculations we used Voight profile and neglected line mixing and other fine effects that may affect line shape in its far wings (Boone et al., 2011). Although such a simplistic approach may appear insufficient for analysis of a fully resolved line profile, experiments presented below show that the model fits observations within data errors. Peculiar exceptions, including poor fitting to the Doppler-broadened line core described below, do not affect wind retrievals as the contribution of this

narrow spectral range to the retrieval procedure is negligible

The modelled atmosphere includes 100 uniform layers with constant gas composition, pressure, temperature, and wind. Upper boundary of the modelled atmosphere is determined by the condition that maximal absorption in the upper layer did not exceed 0.1% of the total absorption. The example of line shape fit by radiative transfer model involving Doppler line distortion is presented in Figure 2 (c) by red curve. It is evident that line shape in the atmospheric column shown in Figure 2

(c) is different from that detected in the reference cell (Figure 2, a). Atmospheric line reveals a sharper tip and broader wings due to superposition of absorption characterized by different broadening at different altitudes. Broader wings dominated by collisional Lorentz profile are formed in the troposphere, whereas narrow, Doppler-broadened tip is formed in the stratosphere and mesosphere. It is this line shape, that preserves information about vertical variations of atmospheric conditions, that provides a clue to wind profile retrievals from LHS data.

Wind affects the model spectrum by Doppler shift of the line at each particular model layer, with line shape and absorption value corresponding to the local pressure, temperature and $CO_2$ concentration. Indeed, lower atmospheric layers contribute more effectively to the wings of line profile, whereas line peak dominated with Doppler thermal broadening is formed at higher altitudes. Therefore, providing non-uniform profile of wind projection on the line of sight, the observed line profile is distorted rather than shifted in frequency scale. This fact gives an opportunity of wind profile retrieval based on analysis of

the resolved line shape.





### 3.2 Solution to the inverse problem of wind profiling based on high resolution NIR absorption spectrum

Inverse problems of radiative transfer similar to those stated above, i.e. wind projection profile retrievals from absorption line shape observed through the atmospheric column suffers from ill-posedness. This term means that a solution is not unique and/or reveals instability versus small variations of data such a solution is based on (Rodgers, 2000). Since TDLHS

introduces certain noise level to data by nature, noisy data forces instability of a solution unless special measures for solution regularization is taken. In this work we consider generalized residual method of regularization (GRM), similar in many aspects to Tikhonov regularization, yet highly efficient in cases when some *a priori* assumption on solution properties could be made (Tikhonov, 1998).

Let us consider a radiative transfer model taking into account Doppler distortion of the absorption spectrum:

$$ \int_0^H \sigma\left(\nu - \frac{\nu_0 \sin(\theta)}{c} U(z), z\right) \rho(z) dz \quad = \quad \tau(\nu), \tag{4}$$

where $\tau(\nu) = -\ln T(\nu) \cdot \cos\theta$ is optical depth normalized for vertical path at wavenumber $\nu$, $T(\nu)$ is observed atmospheric transmission, $\theta$ is the Sun zenith angle, $\rho(z)$ is assumed number density of $CO_2$ molecules, $U(z)$ is assumed vertical profile of wind projection on the line of sight, $c$ is the speed of light, $\nu_0$ - is a wavenumber of the line centre and the function $\sigma(\nu, z)$ is a model absorption crossection at wavenumber $\nu$ and altitude $z$ per molecule calculated according to the

algorithm described above. Equation (4) may be rewritten in the operator form:

$$ \hat{A}\{\overline{U}, \overline{\rho}\} = \overline{\tau}; \left(\overline{U}, \overline{\rho}\right) \in W_2^1; \tau \in L_2, \tag{5}$$

or, assuming a nearly uniform vertical distribution of carbon dioxide in the atmospheric column and replacing vector $\overline{\rho}$ by a scalar parameter $\rho$,

$$ \hat{A}\overline{U} \cdot \rho = \overline{\tau}; \rho \in R; \overline{U} \in W_2^1; \overline{\tau} \in L_2. \tag{6}$$

It has been first noticed by Tikhonov (1943) that a solution $\overline{U}$ to (6) reveals stability in RHS provided it is searched for on a compact set $M \subset W_2^1$. GRM implies introduction of a stabilizing functional $\Omega(U, \rho)$ which is physically sensible and obeys special properties, e.g. is concave and semi-continuous, to provide method convergence (Amato and Hughes, 1991). Such a functional may be constructed according to the maximum entropy method:

$$ \Omega_\varepsilon(U) = \int_0^H g(U(z)) \ln(g(U(z)) + 1) dz \quad + \quad \int_0^H g(\partial U_z(z)) \ln\left(g(\partial U_z(z)) + 1|\right) dz, \tag{7}$$





Function $g(U)$ is introduced to provide convexity of the functional $\Omega_\varepsilon(U)$, $g(U)$ is a smooth function taken in the form

$g(U) = \sqrt{U^2 + \varepsilon}$ with a small epsilon $\varepsilon = 0.001$ and the approximation property $|\Omega_\varepsilon(U) - \Omega_\varepsilon(U)| < \varepsilon k$ ($k$ - is a

constant). So that the solution should be searched for on the compact set $M = \{\Omega(\overline{U}, \overline{\rho}) \leq \Omega_\delta\}$. As a result, the solution is

obtained by constrained minimization of the stabilizing functional provided that model deviation from the dataset does not

exceed expected measurement error:

$$\overline{U}' = \arg\min\left\{\Omega_\varepsilon(\overline{U}) : \left\|\hat{A}\overline{U} \cdot \rho - \overline{\tau}\right\| \leq \delta\right\},$$ (8)

The choice of maximal entropy as a solution regularization criterion is closely related to the nature of its uncertainty. Indeed,

vertical wind profile from the ground to mesosphere is determined by a number of factors and therefore is not expected to

obey any special properties but smoothness caused by averaging over kernels shown in Figure 3. However, it is known *a*

*priori* that the retrievals are based on observed line shapes (see Figure 2b), and may not contain information exceeding its

limited amounts delivered by those data. Any extra information added during data treatment process may only decrease the

quality of retrievals. Therefore, among solutions compatible with observations, those obeying minimal information would be

considered the best. Since information is by definition equivalent to entropy up to a sign, smoothing functional (7) reflecting

maximal entropy principle selects the best solution to the inverse problem (6). Once the regularization criterion is selected,

the problem (8) could be efficiently solved sequential quadratic programming methods (Spellucci, 1998), resulting in vertical

profile of wind projection on the line of sight of Sun observation.

In order to estimate potential uncertainties introduced by the retrievals, it is instructive to linearize the problem (6) in the

proximity of some assumed solution $\overline{U}_0$ :

$$\hat{A}' d\overline{U} \cdot \rho = d\overline{\tau},$$ (9)

where $\hat{A}'$ is an operator of the initial radiative transfer problem (6) linearized in wind velocity projection $\overline{U}$. It is

convenient to use classical Tikhonov functional which is the nonconstrained analog of the GRM method with the Lagrange

additional parameter α:

$$M^\alpha[U] = \left\|\hat{A}\overline{U} \cdot \rho - \overline{\tau}\right\|^2 + \alpha\Omega_\varepsilon(\overline{U}),$$ (10)

Then the Jacobian of the regularized inverse operator may be written in a matrix form, where $J$ is a Jacobian matrix of the

subintegral function in (7):

$$\hat{K} = \left(\hat{A}'^*\hat{A}' + \alpha I\right)\hat{A}'^*,$$ (11)

Where $\alpha$ is a regularization parameter which can be determined according to the residual principle (ref). Here is used that the

function $\rho(\alpha)$ is monotonically increasing according to the $\alpha$ parameter $\rho(\alpha) = \|AU - \tau\|^2 = \delta^2$, in our case we have





$\alpha = 2.43e - 5$. The linearized solution will take a form $\overline{U}' = \rho^{-1} \hat{K} \left( \overline{\tau} - \overline{\tau}_0 \right) + \overline{U}_0$, or $d\overline{U} = \rho^{-1} \hat{K} \, d\overline{\tau}$. Averaging

kernel which characterizes the sensitivity of a regularized solution $\overline{U}'$ to exact one $\overline{U}$ is defined as:

$$\hat{\kappa} = \frac{\partial \overline{U}'}{\partial \overline{U}} = \hat{K} \hat{A}'. \tag{12}$$

Examples of averaging kernels are presented in Figure 4 depending on altitude. Each curve corresponds to particular altitude where exact solution is defined, so that one may considered then as point spread functions associated with the regularization procedure. Therefore, a characteristic width of main peaks means the effective vertical resolution of the method. It is seen from Figure 4, that the resolution is higher in the lower part of the profile and reaches few km near the ground, whereas higher in altitude, it is comparable with one scale height. Another important feature of the kernel is the presence of negative secondary peaks, which may, for some particular wind profiles, generate unrealistic oscillations in retrievals. Nevertheless, the shape of averaging kernels that demonstrate sharp main peaks and regular structure, provides an opportunity of sensible recovery of wind projection profiles from the lower troposphere to mesosphere with the vertical resolution better than one scale height.

**3.3 Results**

Analysis of data received during three observation sessions in summer 2018 is presented below. Retrievals according to the algorithm described in the previous section have been compared with the atmospheric circulation model reanalysis data available in the NCAR/NCEP database (Kalnay et al., 1996). Thermal profiles based on the NCAP reanalysis were adopted for the radiative transfer calculations as well. Reanalysis data with a horizontal resolution of 1 km and 37 layers in the vertical were interpolated to the location of the instrument noticed above, while their lateral variations along the line of sight during Sun observations were neglected.

Wind projection profiles on the Sun observation line of sight are presented in Figure 5 versus reanalysis data. Measured transmission spectra with best-fit radiative transfer simulations are shown at the left panels on the same Figure as well. It is evident that in the case of relatively smooth wind profiles (panels a – f) the retrievals demonstrate remarkable coincidence with reanalysis data in the troposphere and reasonable agreement at higher altitudes. Although the maximal disagreement between two profiles may reach 10 m/s, as seen, for instance, in Figure 5 (a,b), the altitudes where wind projection slews into opposite direction are reproduced in most observations, with the exception of few cases (Figure 5 g,j). Sensitivity of the proposed technique is evidently insufficient to detect wind variations less than 3 m/s, which is particularly evident in the profiles corresponding to low wind speeds (Figure 5 l,n). This sensitivity limit is consistent with estimated stability of the local oscillator at the level of 1MHz (Rodin et al., 2014) and exceeds the estimate based on spectral resolution of 10 MHz. It worth noticing that variations in the absorption spectrum associated with 1 MHz Doppler shift are two orders of magnitude smaller than the width of the Doppler-broadened line tip, and cannot be distinguished in the graph by naked eye. In some



spectra, characterized by remarkable difference between measured and model data (e.g. Figure 5 g,j) the retrievals reveal substantial deviation from reanalysis as well. We hypothesize that in these cases the measurements are contaminated with

instrumental errors, perhaps, caused by electromagnetic influence of electronic equipment, so that the retrieval algorithm failed to reproduce those data with any reasonable wind profile. In one particular case resented in Figure 5 (n), the model fails to reproduce the very tip of $CO_2$ absorption line with the residual being far beyond noise level. The reason of such a deviation is not clear, perhaps, it is concerned with relatively high zenith angle and, hence, airmass factor, so that the approximation of plane-parallel atmosphere in the model calculation is no longer applicable.

Interpretation of the obtained wind projection profiles are beyond the topic of this paper. We also do not analyze retrieved amounts of carbon dioxide in the atmosphere, both column abundance and vertical distribution, which may also be inferred from high resolution NIR spectroradiometry data. These topics will be considered elsewhere. Here we only consider a potential possibility of remote wind sounding by means of simple, portable, and easy-to-use instrument.

## 4 Conclusion

We demonstrated a new method of wind speed remote sensing based on high resolution heterodyne NIR spectroradiometry in the mode of direct Sun observations. A laser heterodyne spectroradiometer using DFB laser as a local oscillator and multipass ICOS reference cell, allows one to measure atmospheric absorption spectrum with spectral resolution $\lambda/\delta\lambda \sim 10^7$-$10^8$ and to retrieve vertical profiles of wind speed projection on the line of sight with the accuracy of 3-5 m/s and vertical resolution varying from 2 km near the ground to 6 km in the stratosphere. The method provides accurate evaluation of wind

speed shear and altitude of its reversal. Profiles characterized with complex structure and multiple reversals are reconstructed with worse accuracy.

**Acknowledgements**

This work has been supported by the Russian Foundation for Basic Research grants # 18-29-24204 (M. V. Spiridonov, I.Sh. Gazizov) and # 19-32-90276 (A.V. Rodin, S.G. Zenevich).

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





**Table 1: Observations of CO2 absorption line profile.**

| Date | Time | Zenith angle, degree | S/N | Position in Figure 5 |
|---|---|---|---|---|
| 11.07.2017 | 11:39 | 35 | 96 | a |
| 11.07.2017 | 18:12 | 69 | 146 | b |
| 02.08.2017 | 9:11 | 54 | 154 | c |
| 02.08.2017 | 9:40 | 50 | 193 | d |
| 02.08.2017 | 9:59 | 48 | 118 | e |
| 02.08.2017 | 10:22 | 46 | 101 | f |
| 02.08.2017 | 10:42 | 44 | 101 | g |
| 02.08.2017 | 11:03 | 42 | 85 | h |
| 02.08.2017 | 11:23 | 40 | 74 | i |
| 02.08.2017 | 11:43 | 39 | 94 | j |
| 02.08.2017 | 12:04 | 38 | 84 | k |
| 02.08.2017 | 12:25 | 38 | 176 | l |
| 02.08.2017 | 12:45 | 38 | 110 | m |
| 31.08.2017 | 12:43 | 47 | 127 | n |





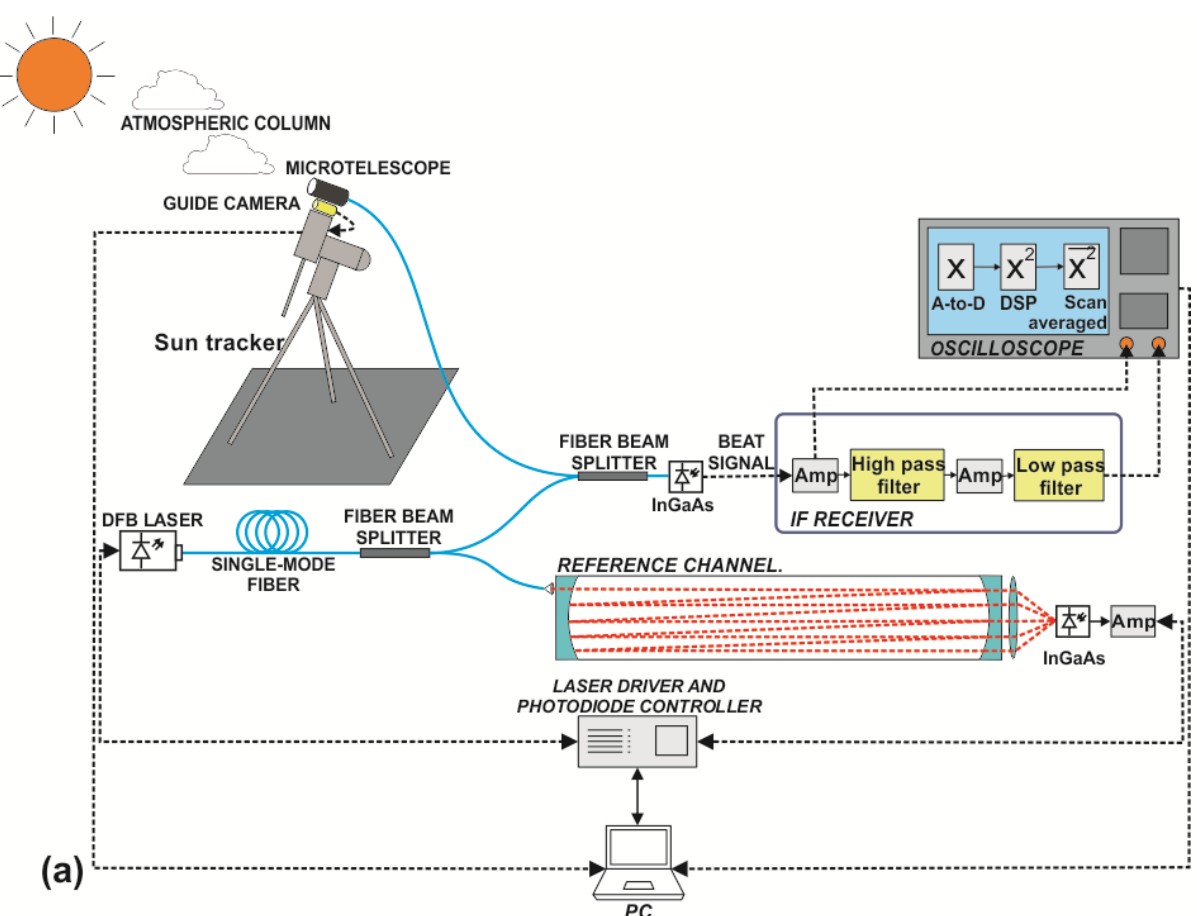


**Figure 1: The sketch of laser heterodyne spectrometer used for wind profiling.**

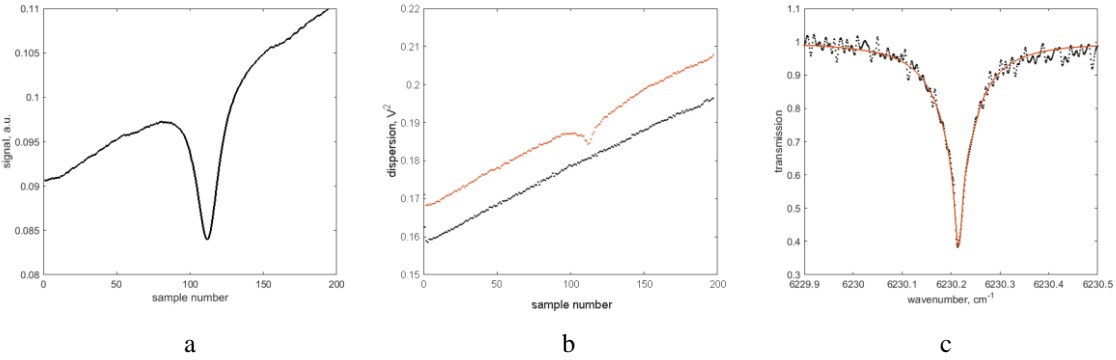

|  |  |  |
|---|---|---|
| a | b | c |

**Figure 2: Heterodyne signal treatment in the LHS: (a) signal in the reference channel during one cycle of laser pumping current ramping; (b) noise dispersion in the heterodyne channel during one cycle of laser pumping current ramping; blue dots – with the Sun off, red dots – with the Sun on; (c) atmospheric transmission spectrum obtained**





**after removing dark signal, normalization to suggested continuum (baseline), and frequency calibration. Black dots – experiment, red curve – model fit.**


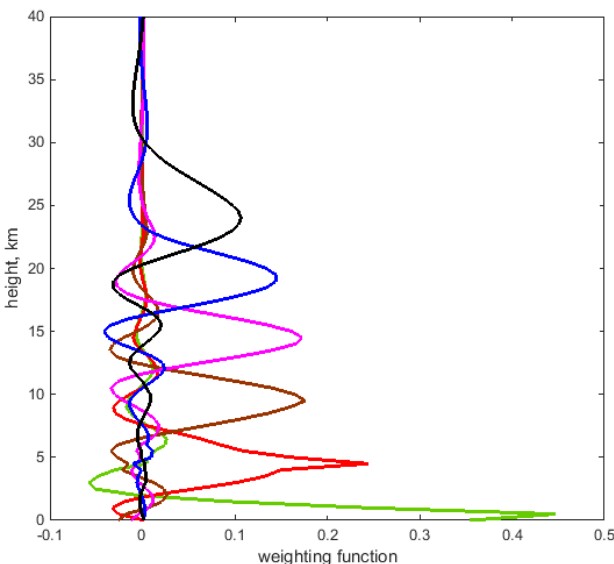

**Figure 4: Weighting kernels of wind retrievals according to Eq. (10) as functions of altitude. Kernels corresponding to reference heights of 0, 5, 10, 15, 20, and 25 km are presented.**

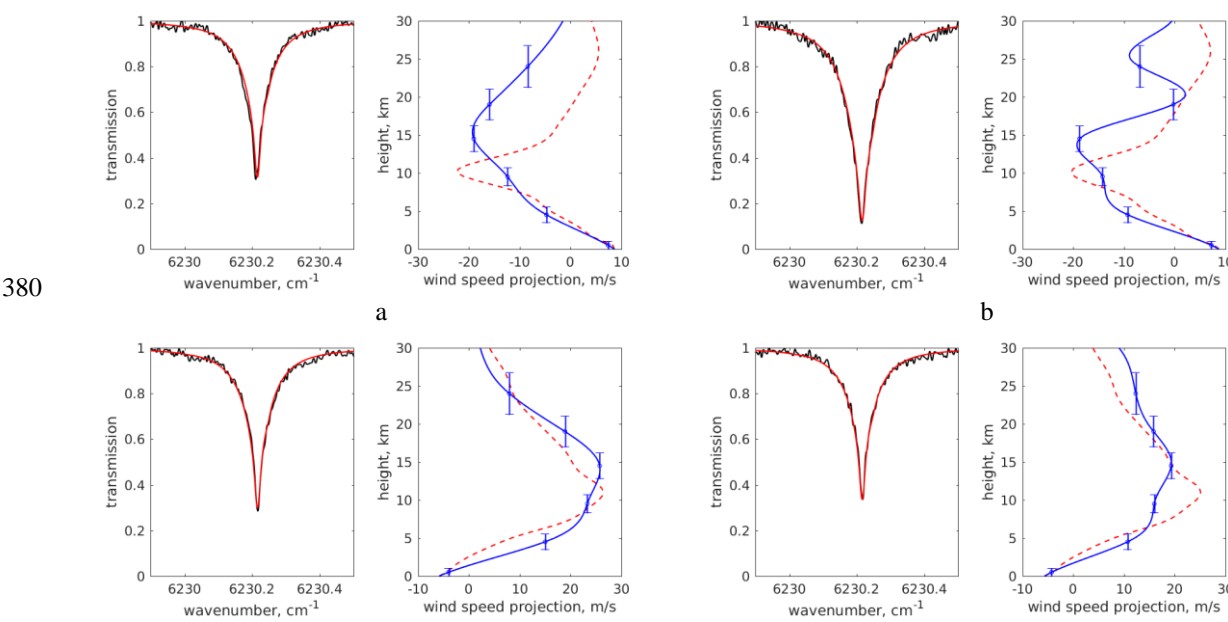






c

d


e

f

g

h

i

j


k

l





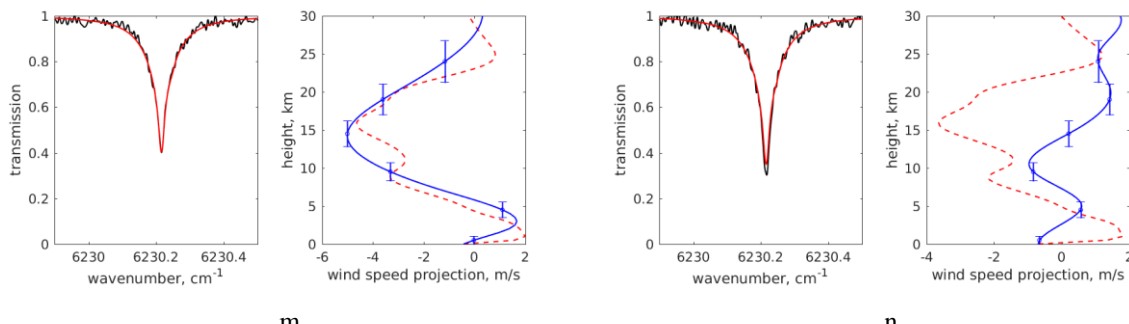

m

n

**Figure 5: Comparison of retrieved wind projection on the line of sight (right panel, blue curve) with reanalysis data (red dashed curve). Vertical error bars correspond to FWHM of a retrieval averaging kernel presented in Figure 4. Left panel shows atmospheric transmission spectrum measured by means of heterodyne technique (black curve) and corresponding best-fit model spectrum (red curve). Date and local time of each observation are presented in Table 1.**