# Peer review of "Vertical wind profiling from the troposphere to the lower mesosphere based on high resolution heterodyne near-infrared spectroradiometry"

_Atmospheric Measurement Techniques, 2019_

## Referee Comment (RC1)

**Review report**

Journal : Atmospheric Measurement Techniques (AMT)

MS No. : amt-2019-444

Title : **Vertical wind profiling from troposphere to the lower mesosphere based on high resolution heterodyne near-infrared spectroradiometry**

Author(s) : Alexander V. Rodin et al.

The authors report on a new technique of wind-speed remote sensing through Doppler shift analysis of a $CO_2$ absorption line at 1.605 μm using high-resolution laser heterodyne spectroradiometry (LHS). A DFB laser-based LHS, combined with a reference cell involving integrated cavity output spectroscopy (ICOS), was used to measure atmospheric absorption spectrum of $CO_2$ with spectral resolution $\lambda/\delta\lambda \sim 10^7 - 10^8$ which allowed to retrieve vertical profiles of wind speed with an accuracy of 3-5 m/s and vertical resolutions varying from 2 km near the ground to 6 km in the stratosphere.

**General comments**

The manuscript is well structured and written, scientifically sound. It would be acceptable for publication in AMT after minor revision by addressing the comments and questions listed below.

**Specific comments**

(1) Spectral resolution issue

The authors emphasize a high spectral resolution of $\lambda/\delta\lambda \sim 10^7 - 10^8$, while once the corresponding laser line width is less than the electronic filter bandwidth, the LHS spectral resolution is determined by the used electronic filter bandwidth. The authors never discuss on the real spectral resolution : what was the used electronic bandwidth and what was its impact on the spatial (vertical) resolution obtained in the retrieved vertical profiles of wind speed? Please make a detailed discussion.

(2) Experimental spectral comparison between LHS and ICOS spectra

The measurement technique presented in the paper relies on Doppler shift analysis of a measured LHR spectrum line related to the same absorption line recorded in a reference cell. It should be important to show an experimental LHS spectral line in comparison with the

reference spectrum and make some discussion because this is the key technical element to support this paper.

(3) Please provide more detailed information on the used devices in the LHS setup, such as laser power, model, etc.

**Technical corrections**

(1) Page 1, line 13 : "provides" should be "provide"

(2) Page 2, line 57 : recent work of Wang et al. on LHR-$CH_4$, in Opt. Ex. **27** (2019) 9610-9619, should be included.

(3) Page 4, line 108 : please check the English usage in the following sentence "single mode fiber geometric aperture factor ~$2.6 \cdot 10^{-8}$ $cm^2$, corresponding to field of view ~0.006°, is close to the maximum available for heterodyne detection expressed by ….?"

(4) Page 5, line 127 : "is passed through consecutive low-pass filter, another amplifier and high-pass filter" should be "is passed through consecutive high-pass filter, another amplifier and low-pass filter" according to Figure 1?

(5) Page 5, line 129 : it would be better to replace "after the filter chain" by "after the IF receiver" according to Figure 1?

(6) Page 6, line 155 : "The ratio of heterodyne signal with subtracted dark signal and baseline approximated by square polynomial ….", the sentence is not clearly stated;

(7) Page 6, line 162 : "In addition to the target $CO_2$ line R2 $14^01\leftarrow00^00$ at 6230.22 $cm^{-1}$, other $CO_2$ lines at 6230.25 $cm^{-1}$, 6230.02 $cm^{-1}$ and 6229.98 $cm^{-1}$ have also been included in calculations" : under atmospheric pressure, how is presented the $CO_2$ line at 6230.25 $cm^{-1}$ related to the line at 6230.22 $cm^{-1}$? It would be better to show the simulation spectrum in combination with the measured LHR spectrum.

(8) Page 6, lines 165 : Please provide more detailed information on the constructed "model of the atmospheric transmission spectrum".

(9) Page 7, line 198 : remove "-" after $\nu_0$;

(10) Page 8, line 233 : "where J is ….", where is J in equation (10)?

(11) Page 8, line 236 : please add the reference(s) in "(ref)";

(12) Page 13, Table 1 : please add the used electronic bandwidth

---

## Referee Comment (RC2) · Anonymous Referee #2 · 6 Feb 2020

The paper "Vertical wind profiling from troposphere to the lower mesosphere based on high resolution heterodyne near-infrared spectroradiometry" by A. V. Rodin et al. makes, as far as I can judge, a compelling case for Doppler-shift based velocimetry. Putting a few minor language issues aside, the paper is well written and well organized. The logical structure of the paper is good. Numerous citations suggest that the work is adequately put in the context of existing work; however, I am not a specialist in this field and thus I cannot judge if any relevant literature is missing.

Many technical aspects of the paper are outside my scope of expertise, thus I may have missed some flaws. Hopefully the other reviewer is closer to this research field.

Fields where I lack expertise are heterodyne spectroscopy and maximum entropy regularization methods.

Scientific issues:

In various places of the text it is not clear that only along line-of-sight velocities are measured. E.g., add in line 95 "with along line-of-sight velocities"; line 180: the along-line-of-sight component of the wind...; since the method provides slant wind velocities, the calculation of the vertical wind speed requires knowledge of the horizontal components. Where are these taken from? This should be discussed latest in Section 3.3. (This is my most serious scientific concern).

I 146-147: Not sure if this statement belongs here. I think it would fit in much better with the Conclusion section.

I 167: The Boone reference seems to refer to a specific method to calculate line mixing. What about also including references on the discovery and measurement of the physical effect (e.g., Armstrong (1982), Bulanin et al. (1984), Strow et al. (1986) or Hartmann (1989)? It will certainly not be necessary to include all of these, but I suggest to include at least one historically relevant paper.

@article{ARMSTRONG82,

```
author = "R. L. Armstrong",
```

title = "Line mixing in the  $\sum_{2 \ , nu_2 \ }$  band of {CO}\_2 \,

journal= "Appl. Opt.",

volume = "21",

number = "12",

pages = "2141-2145",

year = "1982" }

AMTD
@article{BULANIN84,

author = "M. O. Bulanin and A. B. Dokuchaev and M. V. Tonkov and N. N. Filippov",

title = "Influence of Line Interference on the Vibration-Rotation Band Shapes",

journal= "J. Quant. Spectrosc. Radiat. Transfer",

volume = "31",

number = "6",

pages = "521-543",

year = "1984" }

@article{STROW86,

```
author = "L. L. Strow and B. M. Gentry",
```

title = "Rotational collisional narrowing in an infrared {CO} $\ Q\$  branch studied with a tunable-diode laser",

journal= "J. Chem. Phys.",

volume = "84",

number = "3",

pages = "1149-1156",

year = "1986" }

@article{HARTMANN89,

author = "J. M. Hartmann",

title = "Measurements and calculations of {CO\$\_2\$} room-temperature high-pressure spectra in the 4.3~ $\$ mu\$m region",
| journal= "J. Chem. Phys.", |
|----------------------------|
| volume = "90",             |
| number = "6",              |
| pages = "2944-2950",       |
| year = "1989" }            |

1171-179: The radiative transfer model seems not to consider refraction. Is this important? How large is the related error?

same place: Is the numerical accuracy obtained from 100 uniform layers good enough? Is the integration of the radiative transfer equation based on arithmetic mean values of pressure and temperature of the upper and lower boundaries, or is a more sophisticated numerical integration scheme used, relying on mass-weighted mean values or any other integration scheme which takes into account that there are systematically more CO2 molecules in the path segment in the lower half of the layer than in that above? I think that this numerical integration issue deserves some discussion.

I. 189: I think that the adequate reference for ill-posedness is Jacques Hadamard: Sur les problèmes aux dérivées partielles et leur signification physique. In: Princeton University Bulletin. Bd. 13, Nr. 4, 1902, ZDB-ID 1282693-5, S. 49–52. Reference to the Rodgers book can be made, e.g., in the context of Eq. (12).

Technical and Language issues: Caveat: I am not a native English speaker. Thus my language-related recommendations should be taken with care.

- I. 1: ... from THE troposphere...
- I. 11: CO2: use subscript "2"
- I. 14: ... provides AN unambiguous ...
- I. 15: between THE offset ... and THE altitude ... where THE respective ...
I. 16: ...retrieve THE vertical ... with the vertical resolution of the retrievals...

I. 18: ... retrieved wind profileS ...

I. 30: Not only provide heterodyne ...

I. 35: ... have been ...

I. 39: Radiometric Doppler [remove "As"]... AN extraordinary...

I. 46: The heterodyne method...

I. 63: I am not sure if the term "experiment" is appropriate here. An experiment is often understood to be an observation under controlled conditions, and this is not typically the case for atmospheric observations. Perhaps better "instrument", "instrumental setup", or "measurement principle" or something similar.

- I. 82: The power spectrum
- I. 88: ..., the heterodyne signal...
- I. 94: ... that the LO linewidth...
- I. 108: ...with A 5:95 ... According to THE antenna...
- I. 115: both THE heterodyne and THE reference...
- I. 118: ... with A pulse length ... A dead time ... and A stepwise...
- I. 119: ...at A pressure ...
- I. 127: ... through A consecutive ...
- I. 128: ... The preamplifier circuit ... to A Rohde...
- I. 130: THE LO pumping ...
- I. 131: THE oscilloscope (or AN oscilloscope)...
I. 139: ... by THE Fabry...

I. 141: have shown an root mean squares deviation ...

I. 151: Once THE heterodyne signal...

I. 152: ... eliminate THE baseline slope... Since THE heterodyne...

I. 155: of THE heterodyne ... and THE baseline ...

I. 155/156: I would suggest to turn around: The final transmission ... is the ratio ...". Or if you do not want to turn the sentence around, you might with to replace "is" with "renders"

I. 157: notice A baseline...

- I. 161: ... LHS THE scattered ... (???) ... the simulation of atmospheric spectra is...
- I. 163: in THE calculations.
- I. 165: ... A model of ... (???)
- I. 166: In THE calculations, we used the Voigt line shape
- I. 172: THE upper boundary ... uppermost layer ....
- I. 173: AN example of THE line shape fit...
- I. 174: ... that THE line shape ...
- I. 175: The atmospheric line...
- I. 177: collisional Lorentz broadening ... whereas A narrow...
- I. 178: that PROVIDES information {???}
- I. 180: ... with OTS line shape and...
- I. 190: introduces A certain ... forces THE instability ...
I. 191: consider THE generalized...

I. 206: implies THE introduction

I. 217: THE vertical wind ...

I. 236: Here a reference seems not to be resolved.

I. 238: THE averaging kernel...

In general the notation seems not to comply with AMT format guidelines. E.g. Matrices and vectors should be bold face. etc.

I. 241: ...to THAT particular altitude where THE ... so that one may consider it as a point ...

I. 243: Therefore, the characteristic width of the main peak of the averaging kernel is a measure of the effective...

---

## Author Comment (AC1) · 2 Mar 2020

We agree with most of remarks and thank the reviewer who helped us to eliminate an important mistake in the text. We present hereafter the corrected sentences with modified text marked by yellow.

**Specific comments**

**(1) Spectral resolution issue**

The authors emphasize a high spectral resolution of  $\lambda/\delta\lambda \sim 107-108$ , whileonce the corresponding laser linewidth is less than the electronic filter bandwidth, the LHS spectral resolution is determined by the used electronic filter bandwidth. The authors never discuss on the real spectral resolution: what was the used electronic bandwidth and what was its impact on the spatial (vertical) resolution obtained in the retrieved vertical profiles of wind speed? Please make a detailed discussion.

Careful analysis of the instrument has shown that the real electronic filter bandpass is 0.2....3 MHz, rather than 10 MHz, which corresponds to spectral resolving power  $\lambda/\delta\lambda = 6 \cdot 10^7$ . However, as the spectral point spreads function of the spectrometer, determined by the convolution of laser emission line with the electronic filter bandpass, is highly stable, it does not limit the accuracy of wind retrieval. This accuracy is mainly determined by the emission line stability, which has been explored separately and found to stay in the limit of 1 MHz. In turn, vertical resolution of wind profile retrieval does not relate to filter bandwidth, being completely determined by the pressure broadening mechanism. The filter parameters are corrected and corresponding sentences are added to the manuscript.

**Lines 12-14:** Heterodyne spectroradiometric measurements of the solar radiation passed through the atmosphere provides an unprecedented spectral resolution up to  $\frac{\lambda}{\delta\lambda} \sim 6 \cdot 10^7$  with a signal-to-noise ratio more than 100.

**Line 51-53:** With the resolved bandwidth B = 3 MHz and reasonable exposure time  $\tau$  up to few minutes, the quantum limit constrains heterodyne detection by a minimal level of spectral brightness of  $p = \frac{hc}{\lambda\sqrt{B\tau}} \approx 10^{-24}$  W/Hz.

**Lines 127-128:** After the transimpedance preamplifier, the signal is passed through consecutive low-pass filter, another amplifier and high-pass filter, that limits its bandwidth to 0.2...3 MHz.

**Lines 94-96:** Taking into account that LO linewidth has an order of 2MHz, the spectral resolution of heterodyne detection is sufficient to measure Doppler shift of the absorption line in the atmosphere due to air mass motion with velocities greater than 3 m/s, provided high LO stability and sufficient accuracy of intermediate frequency (IF) signal analysis.

Line 230: Note that average kernels and, hence, vertical resolution of wind retrievals is determined by collisional linewidth and signal-to-noise ratio rather than on spectral resolution of the instrument, which is excessively high for such retrievals.

**(2) Experimental spectral comparison between LHS and ICOS spectra**

The measurement technique presented in the paper relies on Doppler shift analysis of a measured LHR spectrum line related to the same absorption line recorded in a reference cell. It should be important to show an experimental LHS spectral line in comparison with the reference spectrum and make some discussion because this is the key technical element to support this paper.

ICOS reference cell has been only employed to stabilize LO, and ICOS line shape does not affect neither spectral resolution nor retrieval procedure. The only useful information from the reference channel is a position of ICOS line peak vs. laser pump current. The shape of reference line is presented in Figure 2(a). Corresponding discussion is added to the text.

Line 125-127: Although line shape in the ICOS cell is different from the Voigt profile, it does not affect LO stabilization procedure, as the only information used in the feedback concerns with the peak position.

(3) Please provide more detailed information on the used devices in the LHS setup, such as laser power, model, etc.

As the description of the LHS setup has been already published in maximal detail in Zenevich et al., 2019, here we focus on the measurement method and wind retrieval technique.

Technical corrections

Со всеми правками соглашаемся?

(1) Page 1, line 13: "provides" should be "provide"

Corrected

(2) Page 2, line 57: recent work of Wang et al.on LHR-CH4, in Opt. Ex. 27(2019) 9610-9619, should be included.

**Done**

(3) Page 4, line 108 : please check the English usage in the following sentence "single mode fiber geometric aperture factor  $\sim 2.6 \cdot 10$ -8cm2, corresponding to field of view  $\sim 0.006^{\circ}$ , is close to the maximum available for heterodyne detection expressed by....?"

The awkward phrase is broken by two:

According to antenna theorem (Siegman, 1968), single mode fiber geometric aperture factor  $\sim 2.6 \cdot 10^{-8}$  cm2, which corresponds to field of view  $\sim 0.006^{\circ}$ . This is close to the maximum available for heterodyne detection and therefore does not limit the instrument sensitivity (Rodin et al., 2014).

(4) Page 5, line127: "is passed through consecutive low-pass filter, another amplifier and high-passfilter" should be "is passed through consecutive high-pass filter, another amplifier and low-passfilter" according to Figure 1?

In fact, the error is in Figure 1 rather than in the description. The instrument scheme is corrected

(5) Page 5, line129: it would be better to replace "after the filter chain" by "after the IF receiver" according to Figure 1?

Corrected

Preamplifier circuit is connected to Rohde & Schwarz RTO 1012 digital oscilloscope at two points: after the IF receiver and just after transimpedance preamplifier.

(6) Page 6, line155: "The ratio of heterodyne signal with subtracted dark signal and baseline approximated by square polynomial....", the sentence is not clearly stated;

Corrected

After subtraction of dark signal, heterodyne signal should be normalized by assumed spectral continuum (baseline) approximated by square polynomial to obtain the final transmission spectrum of the atmosphere.

(7) Page 6, line162: "In addition to the target CO2line R2  $1401 \leftarrow 0000$  at 6230.22 cm-1, other CO2lines at 6230.25cm-1, 6230.02 cm-1and 6229.98 cm-1have also been included in calculations": under atmospheric pressure, how is presented the CO2line at 6230.25cm-1related to the line at 6230.22cm-1?It would be better to show the simulation spectrum in combination with the measured LHR spectrum.

The accounting for additional weak CO2 lines only slightly affect the line shape, so their effect is invisible in the graph. The comparison of simulations with the measured LHR spectra is presented in Figure 2(c).

(8) Page 6, lines 165: Please provide more detailed information on the constructed "model of the atmospheric transmission spectrum".

It is not clear which details could be provided in addition to the presented information on this simplistic model. The number of layers, line list and line shape approximation are provided in the text. To me more clear, a phrase is added:

The model only included gas absorption in the lines mentioned above, whereas scattering processes have been neglected.

(9) Page 7, line198: remove "-"after nu0;

Done

(10) Page 8, line233 : "where J is ....", where is J in equation (10)?

Corrected:

Then the Jacobian of the regularized inverse operator may be written in a matrix form, where I is a Jacobian matrix of the subintegral function in (7):

(11) Page 8, line236: please add the reference(s) in "(ref)";

Done

Where  $\alpha$  is a regularization parameter which can be determined according to the residual principle (Tikhonov, 1998).

(12) Page 13, Table 1 :please add the used electronic bandwidth

In all measurements, electronic bandwidth is 0.2..3 MHz

---

## Author Comment (AC2) · 10 Mar 2020

We agree with most of remarks and thank the reviewer who helped us to eliminate an important mistake in the text. We present hereafter the corrected sentences with modified text marked by yellow.

Scientific issues:

*In various places of the text it is not clear that only along line-of-sight velocities are measured. E.g., add in line 95 "with along line-of-sight velocities"; line 180: the along-line-of-sight component of the wind...; since the method provides slant wind velocities, the calculation of the vertical wind speed requires knowledge of the horizontal components. Where are these taken from? This should be discussed latest in Section 3.3. (This is my most serious scientific concern).*

In fact, any Doppler velocimetry techniques only provides velocity projection on the line of sight, with velocity component normal to the line of sight being inaccessible for Doppler effect. This is a natural limitation of the Doppler method and we cannot pretend to overcome it. However, in meteorology it is commonly accepted that horizontal wind component is typically much higher than the vertical one, due to very strong anisotropy of the atmosphere on the scale of tens km. Hence, with the accuracy of the method we propose, which does not exceed few m/s, we may neglect the vertical wind component and retrieve wind speed projection on the current direction of the Sun. Again, the method is unable to access the normal wind component, if any. Corresponding notes is added to the text, as recommended by the reviewer.

**Line 90:** Taking into account that LO linewidth has an order of 2MHz, the spectral resolution of heterodyne detection is sufficient to measure Doppler shift of the absorption line in the atmosphere due to air mass motion with ==along line-of-sight== velocities greater than 3 m/s, provided high LO stability and sufficient accuracy of intermediate frequency (IF) signal analysis.

**Line 184**: This fact gives an opportunity of ==line-of-sight component of the wind== profile retrieval based on analysis of the resolved line shape.

*l 146-147: Not sure if this statement belongs here. I think it would fit in much better with the Conclusion section.*

Absolutely correct. Done.

**Line 285**: ==Further development of this technique is possible by implementing of multichannel photodetector configuration. This could significantly decrease the integration time and increase signal-to-noise ratio.==

*l 167: The Boone reference seems to refer to a specific method to calculate line mixing. What about also including references on the discovery and measurement of the physical effect (e.g., Armstrong (1982), Bulanin et al. (1984), Strow et al. (1986) orHartmann (1989)? It will certainly not be necessary to include all of these, but I suggest to include at least one historically relevant paper. @article{ARMSTRONG82,*

author = "R. L. Armstrong",

title = "Line mixing in the $\nu_2$ band of {CO$_2$}",

journal= "Appl. Opt.",

volume = "21",

number = "12",

pages = "2141-2145",

year = "1982" }

@article{BULANIN84,

author = "M. O. Bulanin and A. B. Dokuchaev and M. V. Tonkov and N. N. Filippov",

title = "Influence of Line Interference on the Vibration–Rotation Band Shapes",

journal= "J. Quant. Spectrosc. Radiat. Transfer",

volume = "31",

number = "6",

pages = "521-543",

year = "1984" }

@article{STROW86,

author = "L. L. Strow and B. M. Gentry",

title = "Rotational collisional narrowing in an infrared {CO}$_2$ {{\it Q}} branch studied with a tunable diode laser",

journal= "J. Chem. Phys.",

volume = "84",

number = "3",

pages = "1149-1156",

year = "1986" }

@article{HARTMANN89,

author = "J. M. Hartmann",

title = "Measurements and calculations of {CO$_2$} room–temperature high-pressurespectra in the 4.3~$\mu$m region",

journal= "J. Chem. Phys.",

*volume = "90",*

*number = "6",*

*pages = "2944-2950",*

*year = "1989" }*

Reference included

**Line 162**: In calculations we used Voight profile and neglected line mixing and other fine effects that may affect line shape in its far wings (Armstrong, 1982, Bulanin et al., 1984, Boone et al., 2011).

*l171-179: The radiative transfer model seems not to consider refraction. Is this important? How large is the related error?*

This is second-order effect that may be neglected, however, we made it more clear.

**Line 171**: Minor effects of second order in terms of line-of-sight curvative, such as atmospheric refraction and sphericity of the atmosphere have been neglected as well.

*same place: Is the numerical accuracy obtained from 100 uniform layers good enough? Is the integration of the radiative transfer equation based on arithmetic mean values of pressure and temperature of the upper and lower boundaries, or is a more sophisticated numerical integration scheme used, relying on mass-weighted mean values or any other integration scheme which takes into account that there are systematically more CO2 molecules in the path segment in the lower half of the layer than in that above? I think that this numerical integration issue deserves some discussion.*

We employed a standard procedure of synthetic absorption spectrum calculation widely used for atmospheric retrievals. In particular, the model preserves the total amounts of atmospheric gases, which in the hydrostatic approximation is equivalent to the pressure profile. However, even provided some error in the net absorption calculation, wind retrievals are only relevant to the line shape rather than line depth. We added specific note to the text.

**Line 172**: The model does not pretend to provide precise calculation of the net absorption, as if were required for $CO_2$ column amounts retrievals, whereas it is line shape rather than line depth, that contains information about line-of-sight wind component.

*l. 189: I think that the adequate reference for ill-posedness is Jacques Hadamard: Sur les problèmes aux dérivées partielles et leur signification physique. In: Princeton University Bulletin. Bd. 13, Nr. 4, 1902, ZDB-ID 1282693-5, S. 49–52. Reference to the Rodgers book can be made, e.g., in the context of Eq. (12).*

Done:

**Line 189**: This term means that a solution is not unique and/or reveals instability versus small variations of data such a solution is based on (Hadamard, 1902)

**Line 240**: Averaging kernel which characterizes the sensitivity of a regularized solution   to exact one   is defined as (see Rodgers, 2000):

*Technical and Language issues: Caveat: I am not a native English speaker.  Thus mylanguage-related recommendations should be taken with care.*

*l. 1: ... from THE troposphere...l. 11: CO2: use subscript "2"*

Agree

*l. 14: ... provides AN unambiguous ...*

Agree

*l. 15: between THE offset ... and THE altitude ... where THE respective ...*

Agree

*l. 16: ...retrieve THE vertical ... with the vertical resolution of the retrievals...*

Agree

*l. 18: ... retrieved wind profileS ...*

Agree

*l. 30: Not only provide heterodyne ...*

Not only provides….

*l. 35: ... have been ...l. 39: Radiometric Doppler [remove "As"]... AN extraordinary...*

Agree

*l. 46: The heterodyne method...*

Agree

*l.  63:  I am not sure if the term "experiment" is appropriate here.  An experiment is often understood to be an observation under controlled conditions, and this is not typically the case for atmospheric observations. Perhaps better "instrument", "instrumental setup", or "measurement principle" or something similar.*

Changed by

**2 The instrumental setup**

*l. 82: The power spectrum*

Agree

*l. 88: ..., the heterodyne signal...*

Agree

*l. 94: ... that the LO linewidth...*

Agree

*l. 108: ...with A 5:95 ... According to THE antenna...*

Agree

*l. 115: both THE heterodyne and THE reference...*

Agree

*l. 118: ... with A pulse length ... A dead time ... and A stepwise...*

Agree

*l. 119: ...at A pressure ...*

Agree

*l. 127: ... through A consecutive ...*

Agree

*l. 128: ... The preamplifier circuit ... to A Rohde...*

Agree

*l. 130: THE LO pumping ...*

Agree

*l. 131: THE oscilloscope (or AN oscilloscope)...*

Agree

*l. 139: ... by THE Fabry...*

a Fabry…

*l. 141: have shown an root mean squares deviation ...*

Agree

*l. 151: Once THE heterodyne signal...*

Agree

*l. 152: ... eliminate THE baseline slope... Since THE heterodyne...*

Agree

*l. 155: of THE heterodyne ... and THE baseline...l. 155/156: I would suggest to turn around: The final transmission ... is the ratio ...".Or if you do not want to turn the sentence around, you might with to replace "is" with "renders"*

This sentence is rephrased according to another reviewer's comment.

==After subtraction of dark signal, heterodyne signal should be normalized by assumed spectral continuum (baseline) approximated by square polynomial to obtain the final transmission spectrum of the atmosphere.==

*l. 157: notice A baseline...*

Agree

*l. 161: ... LHS THE scattered ... (???) ...the simulation of atmospheric spectra is...*

Agree

*l. 163: in THE calculations.*

Agree

*l. 165: ... A model of ... (???)*

Agree

*l. 166: In THE calculations, we used the Voigt line shape*

Agree

*l. 172: THE upper boundary ... uppermost layer ....*

Agree

*l. 173: AN example of THE line shape fit...*

Agree

*l. 174: ... that THE line shape...*

Agree

*l. 175: The atmospheric line...*

Agree

*l. 177: collisional Lorentz broadening ... whereas A narrow...*

Agree

*l. 178: that PROVIDES information {???}*

Agree

*l. 180: ... with OTS line shape and...*

with transmittance spectrum line shape…

*l. 190: introduces A certain ... forces THE instability ...*

Agree

*l. 191: consider THE generalized...*

Agree

*l. 206: implies THE introduction*

Agree

*l. 217: THE vertical wind ...*

Agree

*l. 236: Here a reference seems not to be resolved.*

Done

*l. 238: THE averaging kernel...In general the notation seems not to comply with AMT format guidelines. E.g. Matrices and vectors should be bold face. etc.*

Agree. Equation format will be corrected later on the editor's demand

*l. 241: ...to THAT particular altitude where THE ...  so that one may consider it as a point ...*

Agree

*l. 243: Therefore, the characteristic width of the main peak of the averaging kernel is a measure of the effective...*

Agree

---

## Author Response (AR2)

Response to the Editor's decision on the manuscript #amt-2019-444

 Dear Colleague,

Thank you for your cooperation. Please find enclosed manuscript with the embedded corrections in *.doc and *.pdf formats. Please note that the funding source has changed, as indicated in the Acknowledgement. The correct grant numbed is 19-29-06104, the list of grantees is the following: A.V. Rodin, M.V.Spiridonov, I.Sh. Gazizov .

Yours sincerely,

Alexander Rodin